# Activated whole-body arginine pathway in high-active mice

Jorge Z. Granados[ID][1,2]\*, Gabriella A. M. Ten Have[2], Ayland C. Letsinger[ID][1], John J. Thaden[2], Marielle P. K. J. Engelen[2], J. Timothy Lightfoot[1‡], Nicolaas E. P. Deutz[2‡]

1 Department of Health and Kinesiology, Biology of Physical Activity Laboratory, Texas A&M University, College Station, TX, United States of America, 2 Department of Health and Kinesiology, Center for Translational Research in Aging & Longevity, Texas A&M University, College Station, TX, United States of America

‡ These authors are joint senior authors on this work.
* jgranados@tamu.edu

**Data Availability Statement:** All relevant data are within the manuscript and its Supporting Information files.

**Funding:** This work was supported by Texas A and M University Research Development Grant from

## Abstract

Our previous studies suggest that physical activity (PA) levels are potentially regulated by endogenous metabolic mechanisms such as the vasodilatory roles of nitric oxide (NO) production *via* the precursor arginine (ARG) and ARG-related pathways. We assessed ARG metabolism and its precursors [citrulline (CIT), glutamine (GLN), glutamate (GLU), ornithine (ORN), and phenylalanine (PHE)] by measuring plasma concentration, whole-body production (WBP), *de novo* ARG and NO production, and clearance rates in previously classified low-active (LA) or high-active (HA) mice. We assessed LA (n = 23) and HA (n = 20) male mice by administering a stable isotope tracer pulse *via* jugular catheterization. We measured plasma enrichments *via* liquid chromatography tandem mass spectrometry (LC-MS/MS) and body compostion by echo-MRI. WBP, clearance rates, and *de novo* ARG and NO were calculated. Compared to LA mice, HA mice had lower plasma concentrations of GLU (71.1%; 36.8 ± 2.9 vs. 17.5 ± 1.7μM; p<0.0001), CIT (21%; 57.3 ± 2.3 vs. 46.4 ± 1.5μM; p = 0.0003), and ORN (40.1%; 55.4 ± 7.3 vs. 36.9 ± 2.6μM; p = 0.0241), but no differences for GLN, PHE, and ARG. However, HA mice had higher estimated NO production ratio (0.64 ± 0.08; p = 0.0197), higher WBP for CIT (21.8%, 8.6 ± 0.2 vs. 10.7 ± 0.3 nmol/g-lbm/min; p<0.0001), ARG (21.4%, 35.0 ± 0.6 vs. 43.4 ± 0.7 nmol/g-lbm/min; p<0.0001), PHE (7.6%, 23.8 ± 0.5 vs. 25.6 ± 0.5 nmol/g-lbm/min; p<0.0100), and lower GLU (78.5%; 9.4 ± 1.1 vs. 4.1 ± 1.6 nmol/g lbm/min; p = 0.0161). We observed no significant differences in WBP for GLN, ORN, PHE, or *de novo* ARG. We concluded that HA mice have an activated whole-body ARG pathway, which may be associated with regulating PA levels *via* increased NO production.

## Introduction

Physical inactivity-related diseases (e.g., cardiovascular ischemic heart disease, diabetes, and colorectal cancer [1]) accounted for ~695,600 of U.S. deaths in 2016 [2] and resulted in an estimated global healthcare cost of $53.8 billion in 2013 [3]. Although moderate physical activity

the Vice-President of Research to JTL. The funder had no role in study design, data collection and analysis, decision to publish, or preparation of the manuscript.

**Competing interests:** The authors have declared that no competing interests exist.

(PA) has been demonstrated to mitigate the incidence of physical inactivity-related diseases [4], fewer than 10% of Americans over the age of 20 adhere to recommended PA guidelines (150 minutes of moderate-intensity per week) [5].

To better understand the potential mechanism(s) regulating PA levels, we have studied inbred high-active (HA) and low-active (LA) mouse models [6–12]. We recently found in HA mice that creatine kinase B and succinyl-CoA ligase are overexpressed in the nucleus accumbens of the brain [13]. Because these two proteins are associated with endogenous metabolism [14], we hypothesize that endogenous metabolism may be involved in the regulation of PA levels.

While there are a variety of endogenous metabolic pathways that could be associated with the regulation of PA levels, we first wanted to study the nitric oxide (NO) precursor, arginine (ARG), and the ARG-related pathways because of NO's known roles in related circulatory pathways [15]. ARG is a conditional essential amino acid (AA) in humans and is derived from: 1) exogenous dietary intake (e.g., nuts, meat products, and nutritional supplements) and serves as a substitute for citrulline (CIT) synthesis through interorgan exchange of ornithine (ORN) conversion within the small intestine *via* arginase II and ornithine transcarbamylase metabolic pathways [16, 17]; 2) whole-body protein breakdown from muscle into phenylalanine (PHE) and glutamine (GLN) [18]; and 3) *via de novo* ARG production within the intestinal-renal axis through CIT catalyzation by the enzymes argininosuccinate synthase and argininosuccinate lyase [18]. ARG is used in many biological functions [19–21], including protein synthesis, creatine synthesis, and NO synthesis [22–25].

Arginine's functions are known to be affected by exercise exposure, particularly the vasodilatory changes associated with NO production [26–30]. Essentially, during PA (e.g., exercise), NO increases blood flow to muscles, thereby increasing delivery of nutrients and clearing of waste products, which may promote longer PA duration [31]. We, therefore, hypothesize that NO derived from ARG may affect PA levels in mouse strains with different inherent PA levels.

To determine if metabolites of the ARG pathways were associated with the regulation of PA levels, we studied total AA concentrations. Additionally, we used a stable tracer approach to assess whole-body production (WBP), and clearance rates of ARG including metabolic precursors (GLN, glutamate (GLU), ornithine (ORN), CIT, and PHE) and products (*de novo* ARG and NO production) in HA and LA inbred mice in order to assess if differences in ARG metabolism were associated with inherent PA levels.

## Materials and methods

### Animals

All procedures were approved by the institutional animal care and use committee (IACUC) of Texas A&M University (IACUC 2015–0159). We assessed a total of 23 male C3H/HeJ mice (inherently LA inbred strain) and 20 male C57L/J mice (inherently HA inbred strain). The inherent activity levels of these two strains are based on our extensive prior observations of activity levels in these mice (average wheel running distance: LA = 0.6 ± 1.1 km/day; HA = 9.5 ± 2.0 km/day) [6, 8–10, 13, 32]. Given the known activity levels of these two mouse strains, and because we have shown that multiple day exposure to running wheels can induce gene expression changes due to exercise exposure [11], we studied naive animals for this study (i.e., animals not exposed to a running wheel). We purchased mice from The Jackson Laboratory (Bar Harbor, ME, USA) at 10-weeks of age and group-housed in standard mouse-cages in a light and temperature-controlled housing facility (12-hour light-dark cycle, room temperature 22–24˚C). Water and a standard chow diet (Harlan Labs, Houston TX; 25.2% protein, 4.0% fat, 39.5% carbohydrate, 3.3% crude fiber, 10% neutral fiber, and 9.9% ash) diet were

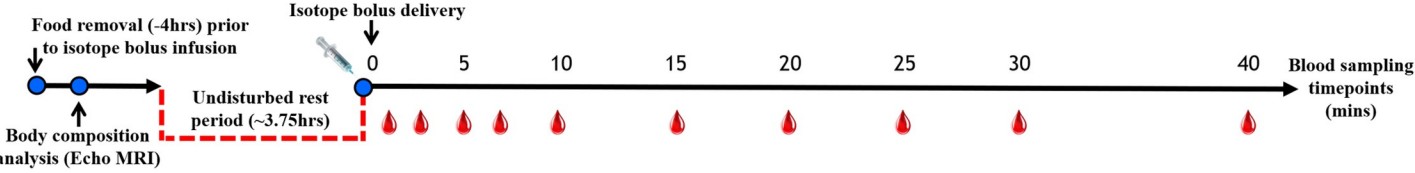

**Fig 1. Study timeline.** Study timeline depicts procedures performed prior to and following isotope bolus delivery (min 0). Blood sampling times are depicted by blood drop images at minutes 1, 3, 5, 7, 10, 15, 20, 30, & 40.

provided *ad libitum*. After a two-week acclimation period, we performed metabolic phenotyping procedures *via* a terminal surgery.

## Study protocol

The experimental protocol was commenced at 8 AM by removing food to study animals in a four-hour post-absorptive condition [33, 34]. Body composition was assessed post food removal, and mice were then placed in clean cages and left undisturbed until the start of the surgical procedures (**Fig 1**). All metabolic testing was performed between 12 PM and 2 PM, using a terminal surgical procedure adapted from Hallemeesch et al. [34], which consisted of sedating the animal and performing a jugular vein catheterization for delivery of isotope tracer bolus and sample collection (**Fig 1**). The study protocol was identical for both HA and LA groups and lasted approximately 5.5 hours.

**Body composition.** Bodyweight (bw) was assessed immediately after food withdrawal using a digital beam scale with lean body mass (lbm), fat mass, percent fat mass, total water, and free water measured *via* echo MRI (EchoMRI LLC, Houston, TX 77079; **Table 1**). Bone mineral density data were collected by dual-energy X-ray absorptiometry (DEXA [Lunar PIXImus densitometer, GE Lunar Corp. Fitchburg, WI]) while the animals were under anesthesia.

**Anesthesia induction.** We anesthetized the mice *via* intraperitoneal (IP) injection (0.1 ml/10g body weight) containing a mixture of medetomidine (2 μg/10g bw) and ketamine (1.25 mg/10g bw), and maintained anesthesia using a continuous pump infusion of medetomidine (0.35 μg/10g bw/h) and ketamine (0.35 mg/ 10 g bw/h) at a rate of 0.1 ml/10 g bw/h, given subcutaneously [34]. We maintained fluid balance and blood pressure by an initial 1.5 ml IP saline injection (0.9% sterile, NaCl), and continuous pump infusion (Harvard PHD2000) of saline at a rate of 2.5 ml/hour delivered subcutaneously [34]. We monitored breathing and core body

**Table 1. Mouse characteristics.**

|  | LA: C3H/HeJ (n = 23) | HA: C57L/J (n = 20) | t-test (p) |
|---|---|---|---|
| Age (weeks) | 12 | 12 | - |
| Body Weight (g) | 25.9 ± 0.3 | 27.5 ± 0.3 | **<0.0001** |
| Lean Mass (g) | 21.1 ± 0.2 | 22.5 ± 0.4 | **0.0003** |
| Fat Mass (g) | 2.6 ± 0.1 | 2.5 ± 0.1 | 0.5255 |
| Free Body Water (g) | 0.024 ± 0.005 | 0.018 ± 0.004 | 0.3398 |
| Total Body Water (g) | 1.79 ± 0.02 | 1.92 ± 0.03 | **0.0006** |
| Bone Mineral Density (g/cm3) | 0.059 ±0.004 | 0.057± 0.005 | 0.0713 |
| Avg. Daily Food Consumption (g) | 3.1 ± 0.2 | 3.4 ± 0.2 | 0.2482 |

Data are mean (±SE) for low-active (LA) and high-active (HA) mice. Statistics are by t-test, bold indicates p<0.05.

temperature ($T_b$) continuously using a rectal thermistor and maintained $T_b$ at a thermoneutral range of 36–37.5˚C *via* heating pad and lamp [35, 36]. A drastic change in $T_b$ can rapidly alter energy and metabolism homeostasis, including metabolic markers assessed in this study [37]. For this reason, we maintained $T_b$ at 36–37.5˚C *via*, while ambient room temperature was maintained at 22–24˚C.

**Stable tracer infusion by IV pulse.** Under anesthesia, a peripheral catheter was placed in the right jugular vein for blood sampling and infusion of a stable isotope tracer pulse (0.1 ml; isotonic) containing L- (Guanidino-$^{15}N_2$) -ARG, L- (5-$^{13}C$; 4,4,5,5-$D_4$) -CIT, L- ($^{13}C_5$) -ORN, L- (1,2-$^{13}C_2$) -GLU, L- ($^{15}N_2$) -GLN, and L- (Ring-$^{13}C_6$) -PHE (Cambridge Isotope Laboratories: Woburn, MA, USA). The different concentrations (nmol/0.01 ml) for each infused isotope tracer are as follows: ARG (381.7), CIT (137.2), ORN (245.9), GLN (1699.6), GLU (196.8), and PHE (271.8).

**Sample collection.** Blood samples (0.05–0.1 ml per sample) were collected utilizing two sampling time schedules (schedule 1: t = 1, 5, 10, 20, and 30 minutes; schedule 2: t = 3, 7, 15, 25, and 40 minutes) after pulse administration (**Fig 1**). Mice were sampled according to schedule 1 or 2 in order to minimize the total volume of blood taken from each animal and to provide a wider range of temporal points for more accurate fitting of the resulting data. Although ~ 2 hours of blood sample collection is common practice in human studies for observing a decay of the administered tracers, our pilot studies show that 30–45 mins of blood sampling is sufficient in mice [38, 39]. The volume of blood that was collected was replaced with an equal volume of sterile normal saline.

In a preliminary study in which no tracer infusion occurred during the mouse surgery, we obtained blood samples to measure the background enrichment. After the cannulation and sampling procedure concluded, the animals were euthanized by removal of the heart. Venous lithium-heparinized blood was collected and immediately placed on ice. Within one hour, the blood samples were centrifuged (4˚C, 3120 x *g* for 5 min) to obtain plasma, which was then deproteinized with 0.1 vol of 33% (w/w) trichloroacetic acid and stored at −80˚C for later analysis of tracer enrichments and concentrations of AAs *via* liquid chromatography tandem-mass spectrometry (LC-MS/MS).

## Biochemical analysis

Plasma AAs and their tracer enrichments were measured batchwise with LC-MS/MS using procedures previously validated in our lab [39–42]. Isotope peak areas were automatically identified and integrated by the SignalFinder1 algorithm in MultiQuant v. 3.0 (Sciex), exported to Excel for calculation of area ratios, and regressed using GraphPad Prism 8.2 as described in detail in our previous study [42]

## Calculations

The decay (change over time) of the tracer/tracee ratio (TTR [injected stable tracers/naturally occurring AA being traced]) was group fitted with a two-exponential least-squares regression: TTR (t) = a*exp(−k1*t) + b*exp(−k2*t). The area under the curve (AUC) was calculated from the integral two exponential curves [43]. Whole-body production (WBP) was then calculated as DOSE (amount of isotope tracer in the pulse)/AUC. Metabolites from the injected stable tracers were group fitted as TTR (t) = −a*exp(−k1*t) + b*exp(−k2*t), and the integral was calculated to represent the AUC.

We calculated the conversion of one AA into another AA by using the WBP of the product AA and the ratio between the AUC of the TTR from the pulse of the product/substrate [42]. For example, the conversion of CIT to ARG (i.e., *de novo* ARG production) was calculated as

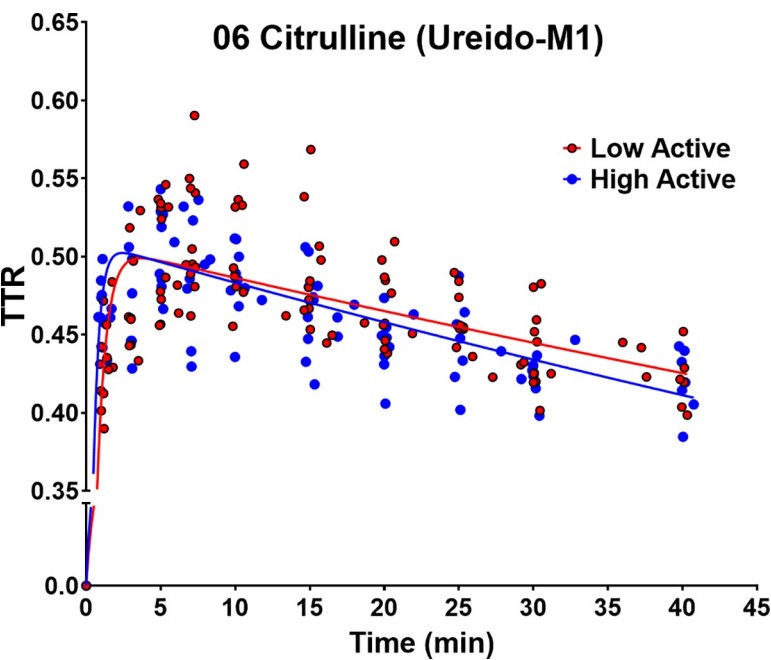

**Fig 2. Logarithmic fitting of 06-citrulline.** Logarithmic fitting of tracer-tracee ratio (TTR) of 06-citrulline [ureido-m1] in low- (LA) and high-active (HA) mice over a 40 min sampling period.

WBP of ARG L-(Guanidino-$^{15}$N$_2$)-arginine $^*$ AUC-TTR (L- [5-$^{13}$C; 4,4,5,5-D$_4$]-arginine/ AUC-TTR L- [5-$^{13}$C; 4,4,5,5-D$_4$]-CIT).

As the AUC calculation with the fitting procedure could not be done for the CIT metabolite tracer (*i.e.*, 06 CIT1 [Ureido-M1]) given that the 40-minute sampling procedure was insufficient to observe differences in the metabolite decay curve (**Fig 2**), based on our previously published work [34, 44–46] we performed an alternative calculation for NO production. As NO production = WBP(CIT) $^*$ AUC (06 CIT1 [Ureido-M1])/AUC(ARG2), it can also be re-written as NO production = WBP(CIT)/AUC(ARG2) $^*$ AUC (06 CIT1 [Ureido-M1]). We subsequently calculated the ratio between the estimated NO production of HA and LA, assuming AUC (06 CIT1 [Ureido-M1]) was not different. The non-compartmental analysis was done using GraphPad Prism (version 8.2). Additionally, the clearance flux of the stable tracers was calculated as WBP/plasma concentration [47] and expressed as mL/min.

### Statistical analysis

Results are expressed as mean ± standard error (SE). If data failed normality or equal variance tests, they were log-transformed. Unpaired Student's *t*-tests were used to determine differences in body composition, plasma AA, and plasma metabolites between the HA and LA mouse groups. Cohen's *d* was then utilized to calculate the effect size between the observed differences between these two strains [48]. A one-sample Wilcoxon *t*-test with a hypothetical value of 1.0 was used to determine differences in NO production ratio. The statistical package within GraphPad Prism (version 8.2) was used for data analysis. The alpha value was set *a priori* at p < 0.05.

### Results

We analyzed a total of 43 male mice at twelve-weeks of age, 23 LA, and 20 HA (**Table 1**). Although no differences in total fat mass, bone mineral density, or average daily food

**Table 2. Plasma amino acid concentrations and clearance rates.**

| Plasma Amino Acid Concentrations (μM) | | | |
|---|---|---|---|
| | LA: (n = 23) | HA: (n = 20) | T-test (p) |
| Glutamate | 36.8 ± 2.9 | 17.5 ± 1.7 | <**0.0001** |
| Glutamine | 641.8 ± 23.9 | 633.7 ± 20.7 | 0.8011 |
| Citrulline | 57.3 ± 2.3 | 46.4 ± 1.5 | **0.0003** |
| Arginine | 104.6 ± 6.7 | 106.7 ± 4.2 | 0.7895 |
| Ornithine | 55.4 ± 7.3 | 36.9 ± 2.6 | **0.0241** |
| Phenylalanine | 86.7 ± 4.3 | 77.5 ± 2.4 | 0.0724 |
| **Plasma Amino Acid Clearance Rate (mL/min)** | | | |
| Glutamate | 0.255 ± 0.036 | 0.234 ± 0.094 | 0.3268 |
| Glutamine | 0.289 ± 0.020 | 0.305 ± 0.059 | 0.2279 |
| Citrulline | 0.150 ± 0.007 | 0.231 ± 0.010 | <**0.0001** |
| Arginine | 0.335 ± 0.022 | 0.407 ± 0.017 | <**0.0001** |
| Ornithine | 0.139 ± 0.020 | 0.228 ± 0.018 | <**0.0001** |
| Phenylalanine | 0.275 ± 0.015 | 0.330 ± 0.012 | <**0.0001** |

Data are mean (±SE). Statistics are by t-test, with bold representing P<0.05.

consumption were observed, the HA mice were characterized by 6.7% higher total body weight (p<0.0001) due to a 6.4% higher lean mass (p = 0.0003). The higher lean mass observed in the HA mice also explains the 7.0% higher total-body water observed in these mice. It should be noted that although the lean mass was higher in the HA mice, it only represents a 1.4 g difference in lean mass between HA and LA mice, suggesting that any differences observed in WBP and clearance rates could be associated with the difference in lean body mass. For this reason, we normalized our results to the animal's lean body mass. The same statement can be applied to the total body water differences as they account for a 1.3 g difference between mouse strains.

Post-absorptive plasma concentrations of the six measured AAs are depicted in **Table 2**. Significantly lower concentrations were observed in the HA mice for GLU (71.1%, p < 0.0001), CIT (21.0%, p = 0.0003), and ORN (40.1%, p = 0.0241), while no significant differences were found in plasma concentrations for GLN, PHE, or ARG.

Despite lower concentrations of GLU, CIT, and ORN in the HA mice, we found that the HA mice had significantly higher WBP for CIT (21.8%, p < 0.0001; **Fig 3A**), while having significantly lower WBP of GLU (78.5%; p = 0.02; **Fig 4A**), with no difference in WBP for ORN (p = 0.17; **Fig 4B**). Additionally, despite no differences in the concentrations of GLN, PHE, or ARG, the HA mice had significantly higher WBP ARG (21.4%, p < 0.0001; **Fig 3B**), and PHE (7.6%, p < 0.01; **Fig 4C**), while having no differences in the WBP of GLN (p = 0.51; **Fig 4D**). Additionally, we found no differences in the conversion of CIT to ARG (i.e., *de novo* ARG production between the LA and HA mice (0.83 ± 0.05 vs. 0.92 ± 0.07 nmol/g lbm/min; p = 0.28).

There were no differences (p = 0.15) during the 40-minute decay curve for the metabolite 06 CIT1 [Ureido-M1] between the groups (**Fig 2**). We then calculated the ratio between estimated NO production of LA (0.79 ± 0.02 nmol/g lbm/min) and HA (1.22 ± 0.03 nmol/g lbm/min) and found that the ratio between the estimated NO production was 0.64 ± 0.08 (p = 0.0197; **Fig 5**). Therefore, the ratio between estimated NO production was lower in LA mice, suggesting HA mice had higher NO production.

With the measurement of AA plasma concentrations and WBP, we calculated the AA clearance rates. We found the HA group had a 42.3% increased clearance rate for CIT, 19.5% for

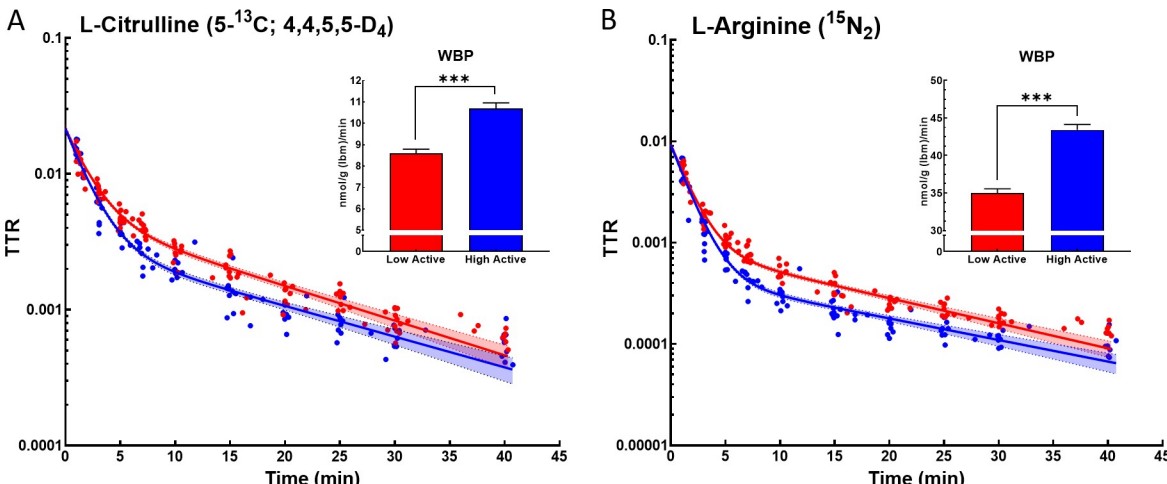

**Fig 3. Citrulline & arginine Whole-Body Production (WBP).** Logarithmic fitting of tracer-tracee ratio (TTR) of (A) L- Citrulline [5-$^{13}$C; 4,4,5,5-D$_4$] and (B) L- Arginine [Guanidino-$^{15}$N$_2$] in low-active (red) and high-active (blue) mice over a period of 40 mins. The fit was utilized for calculations of citrulline and arginine WBP depicted in bar graphs. Data are normalized for lean body mass (lbm) and expressed as mean (± SE). Statistics are by t-test, *** indicates p≤0.001.

# Whole-Body Production

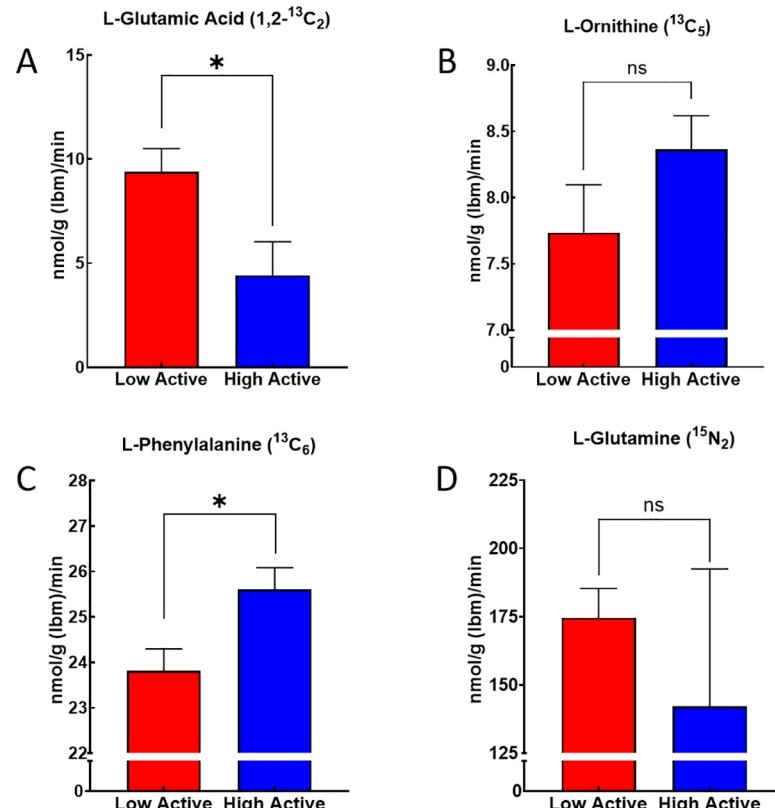

**Fig 4. Whole-Body Production (WBP) of arginine metabolites.** WBP of (A) L- Glutamic Acid [1,2-$^{13}$C$_2$], (B) L- Ornithine [$^{13}$C$_5$], (C) L- Phenylalanine [Ring-$^{13}$C$_6$]), and (D) L- Glutamine [$^{15}$N$_2$] in low- and high-active mice. WBP was calculated from data collected over a 40 min period, normalized for lean body mass (lbm), and expressed as mean (± SE). Statistics are by t-test, * indicates p≤0.05.

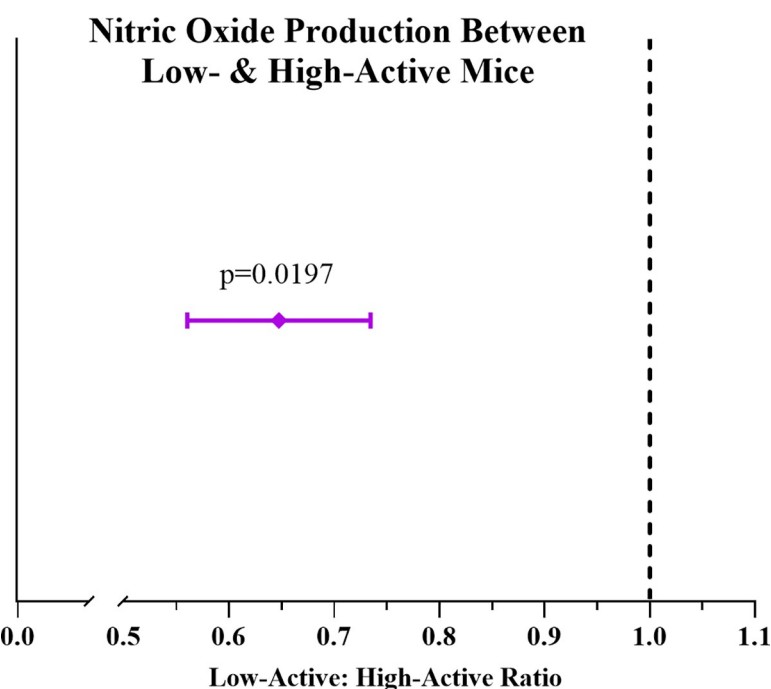

**Fig 5. Nitric Oxide (NO) production ratio.** Dotted line represents a 1:1 ratio in which each mouse strain has equivalent NO production. Purple diamond depicts the ratio between low- (LA) and high-active (HA) mice showing that for every one unit of NO produced by HA mice, the LA mice produce 0.64 units of NO. This suggests HA mice have higher NO production. Data are expressed as mean (95% CI). Statistics are by t-test with alpha level set at p≤0.05.

ARG, and 48.4% for ORN and 18.5% for PHE (all p values < 0.0001) compared to the LA group (**Table 2**). There was no significant difference in clearance rates for GLN (5.3%, p = 0.25) or GLU (8.6%, p = 0.33) in HA mice compared to LA mice (**Table 2**). Overall, the modification of the ARG pathways includes both alterations in whole-body production and clearance of various AA (**Table 3**), which appear to lead to differential NO responses in high active mice.

## Discussion

Despite the HA mice showing higher WBP and clearance capacity of ARG, the lack of change in ARG plasma concentrations between mouse strains can be explained by the high

**Table 3. Summary table.**

| Amino Acid | Baseline Plasma Concentration (umol/l) | Cohen's *d* | WBP (nmol/g lbm/min) | Cohen's *d* | Clearance Rate (WBP/Concentration) | Cohen's *d* |
|---|---|---|---|---|---|---|
| Arginine | No Difference | 0.08 | **HA 21.4% ↑** | **2.79** | **HA 19.5% ↑** | **0.78** |
| Glutamine | No Difference | -0.08 | No Difference | 0.06 | No Difference | 0.08 |
| Glutamate | **HA 71.1% ↓** | **-1.72** | **HA 78.5% ↓** | **-0.84** | No Difference | -0.07 |
| Ornithine | **HA 40.1% ↓** | **-0.71** | No Difference | 0.42 | **HA 48.4% ↑** | **1.01** |
| Citrulline | **HA 21.0% ↓** | **-1.19** | **HA 21.8%↑** | **1.80** | **HA 42.3% ↑** | **2.06** |
| Phenylalanine | No Difference | -0.56 | **HA 7.3% ↑** | **0.78** | **HA 18.5% ↑** | **0.89** |

Data are percent differences and effect size (Cohen's *d*) in high-active (HA) mice compared to low-active mice for amino acid plasma concentration, whole-body production (WBP), and clearance rate. Direction of the arrow depicts a significantly lower or higher value for each amino acid. Cohen's *d* effect size thresholds are: small = 0.2, medium = 0.5, and large = 0.8.

compartmentalization and recycling of ARG within various body organs [18]. For example, the liver produces ARG *via* the complete urea cycle but does not release ARG into plasma, thus not contributing to total ARG plasma concentrations [21]. However, given the compartmentalization of ARG metabolism, alterations in WBP and clearance of ARG may represent other factors that affect endogenous metabolic pathways. For example, it is possible that genomic strain differences could have affected ARG metabolism differentially between the two strains.

To understand the underlying genomic factors that differentially regulate the physical activity levels of HA and LA animals, we have extensively studied these two strains' genomic and proteomic profiles [6, 8, 9, 13, 32]. We found potential proteomic differences in the past [13] that may provide the underlying genomic mechanisms that control the observations we have made. Such proteomic differences include overexpression of succinyl CoA ligase and cluster of creatine kinase B in the nucleus accumbens-brain region that plays a central role in the reward circuit. Interestingly, both succinyl CoA ligase and creatine kinase B are involved in energy metabolism. Primarily, succinyl CoA ligase accelerates the transduction of the intermediate succinyl CoA into the citric acid cycle, and creatine kinase B plays an essential catalytic role in the transfer of phosphate between ATP and several phosphagens within tissues that have significant fluctuating energy demands (e.g., brain, skeletal muscle, heart, and liver).

Moreover, the gene responsible for the metabolic pathway of creatine kinase B is located in chromosome 12 (location: 111669355–111672338) [10], near a single nucleotide polymorphism associated with the regulation of PA distance (location: 89,352,286) [6]. Given that ARG is needed for creatine synthesis [49], creatine is utilized during energy transduction reactions, and our previous study showing overexpression of creatine kinase B in the nucleus accumbens [13], it can be speculated that higher WBP of ARG found in HA mice serves to provide higher energy transduction within skeletal muscle which could be related to their higher PA levels. Therefore, it is probable the differential genomic structure of the HA and LA mice contributed to the differential ARG metabolism observed in this study, a finding that validates using a genomically-controlled model such as inbred mice (versus outbred mice) to explore differential pathways that are associated with physical activity.

In addition to genomic and proteomic factors potentially affecting ARG metabolism, other factors (e.g., age, exposure to running wheel, and diet) may have influenced ARG metabolism pathways in HA and LA mice. Therefore, as a control for aging effects, both mouse groups were analyzed at 12-weeks of age (peak physically active age for most mice [7]). Additionally, while the HA group had higher lean mass than the LA mice, which was different from our previous study [10], but in line with data reported by Reed et al 2007 [50], thus, we controlled for these mass differences by standardizing our results by lean body mass. To prevent potential training-induced changes in metabolism, we studied naive animals (i.e., not exposed to a running wheel) given we have previously shown running wheel exposure can affect gene expression [11]. Lastly, as a check on potential diet-induced changes in metabolism, both strains had the same daily average food consumption, which controlled for potential differences in metabolism induced by varying caloric intake composition or volume. Therefore, other than known genomic differences that have been associated with physical activity regulation, we conclude the animals studied were not exposed to other external factors that would have altered ARG metabolism.

Without external factors altering ARG metabolism, differences in ARG metabolism should be a result of alterations in various endogenous factors. Because ARG can be derived from whole-body protein breakdown, dietary intake, and *de novo* production *via* the intestinal-renal axis [18, 51], we assessed if endogenous factors contributed to the observed WBP of ARG in HA mice. This assessment was supported by two factors: First, given that PHE is a proxy for

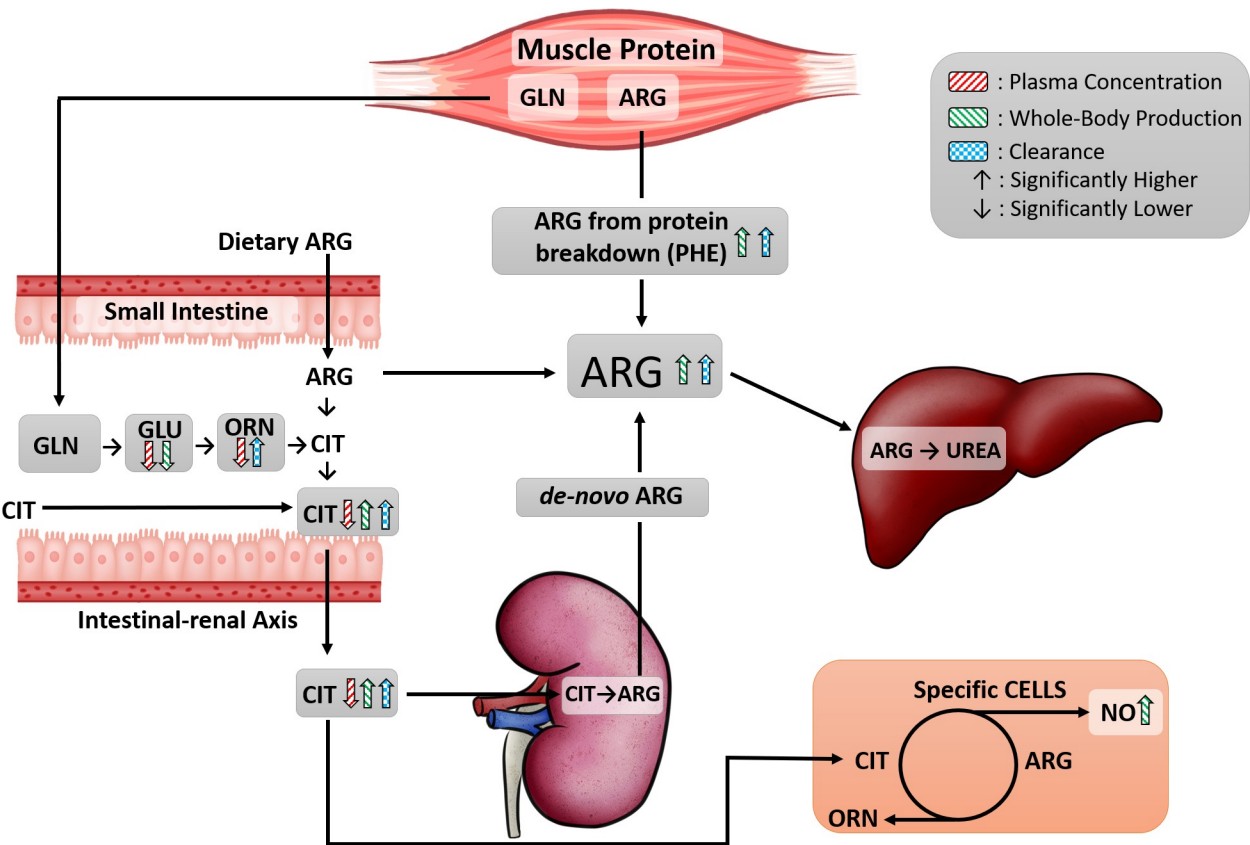

**Fig 6. Arginine metabolic pathway.** Overview of activated whole-body arginine metabolic pathway in high-active mice compared to low-active mice. The direction of arrows depicts significantly higher or lower plasma concentration (red striped), whole-body production (green striped), or clearance (blue checkered) for ARG, CIT, ORN, GLN, GLU, PHE and NO. A lack of arrow within an amino acid or NO signifies no differences between mice.

measuring whole-body protein breakdown [43, 52], the observed higher WBP of PHE suggests that HA mice have higher rates of protein breakdown, which contribute to the higher WBP of ARG. Secondly, because plasma concentrations are associated with the disposal capacity of their corresponding substrates [53], our findings suggest that although the observed lower PHE plasma concentrations were only trending to be significantly lower in the HA mice, this observed lower trend is due to increased clearance rates for PHE. Thus, whole-body protein breakdown and higher clearance capacity of PHE were contributing to the higher WBP of ARG in HA mice independent of exogenous ARG intake (given that diet was controlled).

Another potential source of an increased ARG WBP is through the intestinal-renal axis and the *de novo* ARG production pathways. In order to assess if the higher WBP of ARG is derived from the intestinal-renal axis pathway (**Fig 6**), we assessed plasma concentrations, WBP, and clearance fluxes of CIT, its precursors (GLN, GLU, ORN), and the conversion product of CIT to ARG (i.e., *de novo* ARG) [54]. We found no differences in GLN plasma concentrations, WBP, or clearance rates between strains, suggesting muscle protein breakdown and resynthesis of GLN in the small intestine is constant, and therefore, not affecting GLN as a precursor for CIT production. Additionally, we observed significantly lower plasma concentrations and WBP of GLU in the HA mice without a difference in GLU clearance rate. This observation supports the notion that lower GLU WBP was potentially due to GLU being utilized in other metabolic pathways outside of the small intestine, and thus not contributing to the WBP of

ORN or CIT. The mechanisms responsible for reduced GLU concentrations in the HA animals may include neuronal excitability, synaptic plasticity, immunity, and behavior within the central nervous system [55]; however, we suggest GLU in HA mice is potentially being utilized in an anaplerotic reaction as a substrate to replenish the TCA-Cycle intermediate 2-oxoglutarate when this intermediate is being extracted for biosynthesis.

Also, despite ORN WBP not being different between strains, total plasma concentration was lower, and the clearance rate for ORN was higher in HA mice. This reduction of ORN plasma concentration and increased clearance rate suggest HA mice utilize these AAs at a faster rate than LA mice, which may be related to their higher PA levels. Lastly, we observed decreased CIT plasma concentrations along with an increase in CIT WBP and clearance rate. These observations suggest more CIT is being produced in the small intestine independent of the AA intestinal precursors (i.e., GLN, GLU, ORN) and utilized within the kidney for production of ARG (i.e., *de novo* ARG). However, despite the calculated *de novo* ARG WBP being 10.3% greater in HA mice, this difference was not statistically significant. Therefore, increased WBP and clearance of CIT independent of its precursors and the lack of WBP change in *de novo* ARG suggest the intestinal-renal axis pathway is not responsible for the elevated WBP of ARG in HA mice.

Although ARG is a versatile AA with multiple metabolic fates (e.g., synthesis of protein, creatine, polyamines, agmatine, urea [19]), we hypothesize the higher WBP and clearance of ARG observed were related to higher production of NO which influenced the high activity level of these mice. We base this hypothesis on two observations. First, the combined WBP levels of CIT and PHE contributed to a higher ARG WBP in HA mice. Secondly, the observed higher ratio utilized as a proxy for NO production. Therefore, a higher NO production would increase vasodilation, providing HA mice with increased blood flow, nutrient delivery, and waste removal in the working tissues (e.g., muscles). Consequently, the elevated ARG pathway presents itself as a metabolic mechanism which in theory, could influence the PA levels of HA mice.

## Limitations

It should be noted that the WBP values reported in this study are ~ four-fold higher than what we have previously reported in other studies [34, 56–59]. Our previous rodent studies used primed-constant infusion protocols, which can cause an underestimation of GLN, given that GLN has a considerably large pool size [60]. However, the present study used a single pulse approach, which we expected to provide us with higher values that are probably less than that of the actual values [42, 61].

Moreover, despite our efforts to prevent potential training-induced changes in metabolism by studying naive animals (i.e., not exposed to a running wheel), it is possible that differences in daily cage ambulation could have altered metabolic pathways. However, we have previously shown that running wheel measures of activity do not correlate with measures of daily cage ambulation [10].

## Conclusion

Our observations suggest an activated ARG pathway in those mice that were inherently more physically active. Moreover, the higher ratio for estimation of NO production in HA mice shows the activated ARG pathway may serve as a precursor to increasing NO production, which may be potentially linked to their exhibition of higher PA levels. To obtain a better understanding of how this activated ARG pathway may be linked to higher PA levels in the HA mice, future studies should focus on analyzing keto-acid metabolism along with various organ-tissue analysis (e.g., tissue amino acid concentrations and fractional synthesis rates).

## Supporting information

**S1 File.**
(ZIP)

## Acknowledgments

The authors would like to thank Cristina Osorio, Jeremiah Velasco, and Victor Garcia for their assistance in sample collection and sample preparation for this study. Additionally, we would like to thank Miranda Letsinger for designing the tissue organ graphics shown in **Fig 6**.

## Author Contributions

**Conceptualization:** Gabriella A. M. Ten Have, Marielle P. K. J. Engelen, J. Timothy Lightfoot, Nicolaas E. P. Deutz.

**Data curation:** Jorge Z. Granados, Gabriella A. M. Ten Have, Ayland C. Letsinger, John J. Thaden.

**Formal analysis:** Jorge Z. Granados, Gabriella A. M. Ten Have, John J. Thaden, J. Timothy Lightfoot, Nicolaas E. P. Deutz.

**Funding acquisition:** J. Timothy Lightfoot, Nicolaas E. P. Deutz.

**Investigation:** Jorge Z. Granados, Gabriella A. M. Ten Have, Ayland C. Letsinger, Marielle P. K. J. Engelen, J. Timothy Lightfoot, Nicolaas E. P. Deutz.

**Methodology:** Jorge Z. Granados, Gabriella A. M. Ten Have, Nicolaas E. P. Deutz.

**Project administration:** Jorge Z. Granados, Gabriella A. M. Ten Have.

**Supervision:** Nicolaas E. P. Deutz.

**Validation:** Jorge Z. Granados, Nicolaas E. P. Deutz.

**Visualization:** Jorge Z. Granados.

**Writing – original draft:** Jorge Z. Granados, J. Timothy Lightfoot, Nicolaas E. P. Deutz.

**Writing – review & editing:** Jorge Z. Granados, Gabriella A. M. Ten Have, Ayland C. Letsinger, John J. Thaden, Marielle P. K. J. Engelen, J. Timothy Lightfoot, Nicolaas E. P. Deutz.

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
