## [Decision Letter · Decision Letter 0]

5 Feb 2020

PONE-D-19-33282

Activated whole-body arginine pathway in high-active mice

PLOS ONE

Dear Mr Granados,

Thank you for submitting your manuscript to PLOS ONE. After careful consideration, we feel that it has merit but does not fully meet PLOS ONE’s publication criteria as it currently stands. Therefore, we invite you to submit a revised version of the manuscript that addresses the points raised during the review process. Particularly, you will see that reviewers asked for additional information and for clarification on some aspects of your study.

We would appreciate receiving your revised manuscript by Mar 21 2020 11:59PM. To enhance the reproducibility of your results, we recommend that if applicable you deposit your laboratory protocols in protocols.io, where a protocol can be assigned its own identifier (DOI) such that it can be cited independently in the future. For instructions see: http://journals.plos.org/plosone/s/submission-guidelines#loc-laboratory-protocols

We look forward to receiving your revised manuscript.

Kind regards,

François Blachier, PhD

Academic Editor

PLOS ONE

Journal Requirements:

Reviewers' comments:

Reviewer's Responses to Questions

**Comments to the Author**

1. Is the manuscript technically sound, and do the data support the conclusions?

Reviewer #1: Yes

Reviewer #2: No

Reviewer #3: Yes

2. Has the statistical analysis been performed appropriately and rigorously? 

Reviewer #1: Yes

Reviewer #2: Yes

Reviewer #3: No

3. Have the authors made all data underlying the findings in their manuscript fully available?

Reviewer #1: Yes

Reviewer #2: No

Reviewer #3: Yes

4. Is the manuscript presented in an intelligible fashion and written in standard English?

Reviewer #1: Yes

Reviewer #2: Yes

Reviewer #3: Yes

5. Review Comments to the Author

Reviewer #1: General comments:

This is a good piece of scientific work presented factually. Of optimal length, the paper flows well and highlights relevant areas of knowledge.

This study dealt with the hypothesize that endogenous metabolism may be involved in the regulation of physical activity levels. To determine if metabolites of the arginine pathways were associated with the regulation of physical activity levels, authors studied total amino acid concentrations, using a stable tracer approach to assess whole-body production, and clearance rates of arginine, including metabolic precursors glutamine, glutamate, ornithine, citrulline, and phenylalanine, as well as the products de novo arginine and nitric oxide production in high-active and low-active inbred mice, in order to assess if differences in arginine metabolism were associated with inherent physical activity levels. Overall, the study is relevant and well written. The concept is a good idea and will be a needed addition to the existing body of literature on arginine metabolism.

Specific comments:

- The abstract background is too long. Authors should reduce the background and increase the wording of the results including the numerical values and the observed statistical significance.

- At line 33, describe LC-MS / MS

- The graphic quality of all figures needs to be improved.

Reviewer #2: The manuscript entitled Activated whole-body arginine pathway in high-active mice aimed to assess ARG metabolism and its precursors [citrulline (CIT), glutamine (GLN), glutamate (GLU), ornithine (ORN), and phenylalanine (PHE)] by measuring plasma concentration and enrichments, whole-body production (WBP), de novo ARG and NO production, and clearance rates in mice previously classified as either low-active (LA) or high-active (HA). The authors concluded that HA active mice have an activated whole-body ARG pathway, which may be associated with regulating PA levels via increased NO production.

Major comments:

Abstract

Some data must be included in the abstract to make clear the results.

Methods

A number of the Ethical process is necessary according to COPE;

It is not clear why the authors keep the temperature in 37o C during the surgical procedures since the ideal for mice is about 20-22 o C. Explain!

What time the experimental procedures were carried out?

Please insert the reason why the authors have chosen forty-minutes decay for measurements of plasma AAs and their tracers? A reference regarding the pharmacokinetics of these AAs would be useful.

Results

It is not clear why the HA mice showed a higher lean mass. If they exercised in a wheel-running usually a lower lean mass is expected if they did not, the authors should address this finding properly.

Figure 5 is not clear. Please insert the numbers obtained in a Table or draw a figure to make clear the difference between the groups. Please clarify!

Discussion

First of all, the authors should focus on the discussion of the obtained results of the study and avoiding repeating the results section.

Why plasma arginine is not decreased in HA mice since the primary hypothesis is that Arg/NO pathway is up-regulated in this strain mice? Please clarify!

Minor comments

Please rewrite the discussion section, some phrases do not make sense:

….The basis for the difference between inbred strains is the underlying genomic variations between these strains.

…..Other external factors that could have influenced ARG metabolism pathways were controlled for during this study.

Reviewer #3: This study aimed to evaluate if metabolites of the arginine pathways are associated with the regulation of mice’s physical activity levels.

According to the authors, high active mice presented an activated whole-body arginine pathway, and they suggested that it may be associated with regulating physical activity levels due to higher NO production.

The study is of high quality, is well designed, organized and written. Author’s presented a clear hypothesis, which facilitates the comprehension of the manuscript’s rationale.

The methods used are all well-accepted and described. Results are presented in a logical manner, which facilitates the comprehension of the paper. All finds are very interesting and will clearly contribute to the academic area.

Please find below some suggestions and comments in the attempt to contribute to the quality of the manuscript:

1- The references used in the manuscript are appropriate. However, only 17 from all 54 references used (~31%) are from the last 5 years;

2- Unfortunately, the conversion of the manuscript to the pdf version affected the quality of the figures. They are practically unreadable;

3- As cited before, the manuscript is well written. Only two observations: (1) page 18 line 353; (2) I found the last paragraph of the discussion (conclusion) a little confused;

4- The discussion section should start by presenting the main find of the study, therefore, the entire first phrase could be deleted (repeated information – not necessary);

5- Does this study have any limitations? It would be good to have it described in the discussion. Some “directions” for further studies should be also added;

6- Regarding statistical analysis, why authors preferred to transform data instead use a non-parametric analysis? Lastly, I would suggest authors perform the calculations of the effect size from LA and HA comparisons, and added this information in the referred table.

6. PLOS authors have the option to publish the peer review history of their article (what does this mean?). If published, this will include your full peer review and any attached files.

Reviewer #1: No

Reviewer #2: Yes: Angelina Zanesco

Reviewer #3: Yes: Rafael Herling Lambertucci

---

## [Author Response · Author response to Decision Letter 0]

23 May 2020

Dear editor and reviewers,

We thank the reviewers for their important and valuable time and comments. We have addressed these comments in the revised manuscript with track changes, made the required updates and therefore feel that it has improved the manuscript considerably.

Below are the point-by-point responses to each comment along with supporting citations:

Reviewer #1: General comments:

This is a good piece of scientific work presented factually. Of optimal length, the paper flows well and highlights relevant areas of knowledge.

This study dealt with the hypothesize that endogenous metabolism may be involved in the regulation of physical activity levels. To determine if metabolites of the arginine pathways were associated with the regulation of physical activity levels, authors studied total amino acid concentrations, using a stable tracer approach to assess whole-body production, and clearance rates of arginine, including metabolic precursors glutamine, glutamate, ornithine, citrulline, and phenylalanine, as well as the products de novo arginine and nitric oxide production in high-active and low-active inbred mice, in order to assess if differences in arginine metabolism were associated with inherent physical activity levels. Overall, the study is relevant and well written. The concept is a good idea and will be a needed addition to the existing body of literature on arginine metabolism.

Specific comments:

1- The abstract background is too long. Authors should reduce the background and increase the wording of the results including the numerical values and the observed statistical significance.

Response- We have reduced the abstract background and edited the results section to include numerical values and statistical significance. 

2- At line 33, describe LC-MS / MS

Response- We have described LC-MS/MS in the abstract as suggested.

3- The graphic quality of all figures needs to be improved.

Response – We improved the quality of the figures.

Reviewer #2: The manuscript entitled Activated whole-body arginine pathway in high-active mice aimed to assess ARG metabolism and its precursors [citrulline (CIT), glutamine (GLN), glutamate (GLU), ornithine (ORN), and phenylalanine (PHE)] by measuring plasma concentration and enrichments, whole-body production (WBP), de novo ARG and NO production, and clearance rates in mice previously classified as either low-active (LA) or high-active (HA). The authors concluded that HA active mice have an activated whole-body ARG pathway, which may be associated with regulating PA levels via increased NO production.

Major comments:

Abstract

1- Some data must be included in the abstract to make clear the results.

Response- We have edited the abstract results section to include numerical values and statistical significance. 

Methods

2- A number of the Ethical process is necessary according to COPE;

Response- We have added the approval number provided by the institutional animal care and use committee (IACUC 2015-0159) to the materials and methods section. 

3- It is not clear why the authors keep the temperature in 37o C during the surgical procedures since the ideal for mice is about 20-22 o C. Explain!

Response- We appreciate the reviewer pointing this out as it does need further clarification. We agree that the optimal ambient temperature for mice is 20-22°C; however, we were referring to maintaining the core body temperature at thermoneutral levels; core temperature for mice is ~36.5°C (1) Gordon (1) (2). Maintaining core temperature prevents the alteration of metabolic markers including the ones we assessed in this study (3). For this reason, we maintained the core body temperature between 36 and 37.5°C by using a heating pad and heating lamp as performed in our previous studies (4-6). We have clarified this in the manuscript (anesthesia induction section: Lines 126-130).

4- What time the experimental procedures were carried out?

Response- The experimental procedures began at 8 am (food removal) to start a 4hr post absorptive state (4-6) followed by body composition assessment, with surgical procedure beginning at 12pm - 2pm hours. We have added these times to the study protocol section: Lines 95-98.

5- Please insert the reason why the authors have chosen forty-minutes decay for measurements of plasma AAs and their tracers? A reference regarding the pharmacokinetics of these AAs would be useful.

Response- We have added an explanation along with two references in the sample collection section (lines 144-146). The plasma AA concentration was measured in the blood samples before the pulse of stable tracers was given. We at max needed to collect blood after the pulse for about 45 min to be able to calculate the whole-body production from the decay curve of the enrichment in plasma of the measured amino acids because in pilot experiments, we observed that in mice about 30-45 min is sufficient. In humans, about 2 hours is needed (7, 8).

Results

6- It is not clear why the HA mice showed a higher lean mass. If they exercised in a wheel-running usually a lower lean mass is expected if they did not, the authors should address this finding properly.

Response- It is a reasonable hypothesis that exposure to wheel-running can result in a lower lean mass. However, these mice were not exposed to a running wheel or any other type of exercise activity for this study, instead using naive mice so that the wheel-running exposure did not alter their inherent state (Methods: animals section lines 82-88, and discussion section lines 303-305 where we reference previous work from our lab which has shown this). We have further clarified this topic in the methods. Furthermore, in the discussion section (line 302), we added a citation (9) that shows very similar lean mass results (~1.5g greater lean mass in high active compared to the low active animals. 

7- Figure 5 is not clear. Please insert the numbers obtained in a Table or draw a figure to make clear the difference between the groups. Please clarify!

Response – We have improved the legend of the figure (lines 247-251) to make clear what the figure represents. In brief, as we cannot calculate the actual NO production but only can show the variable part of the calculation of the NO production, we did an estimation based on comparing the variable part of the calculation between the groups (lines 241-243). This LA/HA ratio is shown in the figure with a confidence interval of this ratio. As the LA/HA ratio is significantly lower than 1, we concluded that LA mice have a lower NO production. 

Discussion

8- First of all, the authors should focus on the discussion of the obtained results of the study and avoiding repeating the results section.

Response – We agree with the reviewer and have kept the results in the result section and the discussion in the discussion section and not repeating the results in the discussion section.

9- Why plasma arginine is not decreased in HA mice since the primary hypothesis is that Arg/NO pathway is up-regulated in this strain mice? Please clarify!

Response- We hypothesize that the reason the baseline plasma arginine concentration is not decreased, is because the up-regulated whole-body arginine production is being cleared just as quickly as it is being produced, thus not altering the arginine concentration. The plasma concentration does not relate very well with the production. We have published this observation previously (e.g. (10)).

Minor comments

10- Please rewrite the discussion section, some phrases do not make sense:

….The basis for the difference between inbred strains is the underlying genomic variations between these strains.

Response - We have removed the sentence listed above and rewritten other sentences and phrases which did not make sense in the discussion section.

11-…..Other external factors that could have influenced ARG metabolism pathways were controlled for during this study.

Response – We have rewritten this sentence along with rewriting other parts of the discussion section to make it more legible/easier to follow for the reader. 

Reviewer #3: This study aimed to evaluate if metabolites of the arginine pathways are associated with the regulation of mice’s physical activity levels.

According to the authors, high active mice presented an activated whole-body arginine pathway, and they suggested that it may be associated with regulating physical activity levels due to higher NO production.

The study is of high quality, is well designed, organized and written. Author’s presented a clear hypothesis, which facilitates the comprehension of the manuscript’s rationale.

The methods used are all well-accepted and described. Results are presented in a logical manner, which facilitates the comprehension of the paper. All finds are very interesting and will clearly contribute to the academic area.

Please find below some suggestions and comments in the attempt to contribute to the quality of the manuscript:

1- The references used in the manuscript are appropriate. However, only 17 from all 54 references used (~31%) are from the last 5 years;

Response – We have updated the references to more papers from the last 5 years.

2- Unfortunately, the conversion of the manuscript to the pdf version affected the quality of the figures. They are practically unreadable;

Response – We improved the quality of the figures.

3- As cited before, the manuscript is well written. Only two observations: (1) page 18 line 353 (new version line 363); (2) I found the last paragraph of the discussion (conclusion) a little confused;

Response- (1) We have corrected the misspelling error on line 363.

Response- (2) We have edited the last paragraph of the discussion to increase reader clarity (lines 380-386). 

4- The discussion section should start by presenting the main find of the study, therefore, the entire first phrase could be deleted (repeated information – not necessary);

Response- Thank you for the suggestion. We have deleted the first phrase of the discussion section. 

5- Does this study have any limitations? It would be good to have it described in the discussion. Some “directions” for further studies should be also added;

Response – We included a limitation section along with future directions at the end of the discussion/conclusion section (lines 367-378, and 383-386 respectively).

6- Regarding statistical analysis, why authors preferred to transform data instead use a non-parametric analysis? Lastly, I would suggest authors perform the calculations of the effect size from LA and HA comparisons and added this information in the referred table.

Response – Biostatisticians (11, 12) advise that when there is a lognormal distribution to normalize the data with log transformation and then again test that the distribution is normal. When the distribution is normal, parametric tests can be used. It is always advised to use parametric tests, as non-parametric tests are less powerful.

Effect size calculations: We performed a Cohen’s d test to calculate the effect size from the LA and HA comparisons and added them to the summary table (table 3: line 261) 

1. Gordon CJ. Thermal physiology of laboratory mice: Defining thermoneutrality. 2012;37(8):654-85.

2. Refinetti R. The circadian rhythm of body temperature. Frontiers in Bioscience. 2010;15(1):564.

3. Reitman ML. Of mice and men - environmental temperature, body temperature, and treatment of obesity. FEBS Letters. 2018.

4. Luiking YC, Hallemeesch MM, van de Poll MC, Dejong CH, de Jonge WJ, Lamers WH, et al. Reduced citrulline availability by OTC deficiency in mice is related to reduced nitric oxide production. American journal of physiology Endocrinology and metabolism. 2008;295(6):E1315-22.

5. Luiking YC, Hallemeesch MM, Lamers WH, Deutz NEP. NOS3 is involved in the increased protein and arginine metabolic response in muscle during early endotoxemia in mice (vol 288, pg E1258, 2006). Am J Physiol-Endoc M. 2007;292(1):E369-E.

6. Hallemeesch MM, Ten Have GA, Deutz NE. Metabolic flux measurements across portal drained viscera, liver, kidney and hindquarter in mice. Lab Anim. 2001;35(1):101-10.

7. Engelen MPKJ, Ten Have GAM, Thaden JJ, Deutz NEP. New advances in stable tracer methods to assess whole-body protein and amino acid metabolism. Current opinion in clinical nutrition and metabolic care. 2019;22(5):337-46.

8. Jonker R, Deutz NEP, Harrykissoon R, Zachria AJ, Veley EA, Engelen M. A critical evaluation of the anabolic response after bolus or continuous feeding in COPD and healthy older adults. Clinical science. 2018;132(1):17-31.

9. Reed DR, Bachmanov AA, Tordoff MG. Forty mouse strain survey of body composition. 2007;91(5):593-600.

10. Jonker R, Deutz NE, Erbland ML, Anderson PJ, Engelen MP. Alterations in whole-body arginine metabolism in chronic obstructive pulmonary disease. Am J Clin Nutr. 2016;103(6):1458-64.

11. Motulsky HJ. Common misconceptions about data analysis and statistics. Naunyn-Schmiedeberg's Archives of Pharmacology. 2014;387(11):1017-23.

12. Motulsky HJ. Common misconceptions about data analysis and statistics. British journal of pharmacology. 2015;172(8):2126-32.

---

## [Decision Letter · Decision Letter 1]

9 Jun 2020

Activated whole-body arginine pathway in high-active mice

PONE-D-19-33282R1

Dear Dr. Granados,

We’re pleased to inform you that your manuscript has been judged scientifically suitable for publication and will be formally accepted for publication once it meets all outstanding technical requirements.

Kind regards,

François Blachier, PhD

Academic Editor

PLOS ONE

Additional Editor Comments (optional):

Reviewers' comments:

Reviewer's Responses to Questions

**Comments to the Author**

1. If the authors have adequately addressed your comments raised in a previous round of review and you feel that this manuscript is now acceptable for publication, you may indicate that here to bypass the “Comments to the Author” section, enter your conflict of interest statement in the “Confidential to Editor” section, and submit your "Accept" recommendation.

Reviewer #1: All comments have been addressed

Reviewer #2: All comments have been addressed

Reviewer #3: All comments have been addressed

2. Is the manuscript technically sound, and do the data support the conclusions?

Reviewer #1: Yes

Reviewer #2: Yes

Reviewer #3: Yes

3. Has the statistical analysis been performed appropriately and rigorously? 

Reviewer #1: Yes

Reviewer #2: Yes

Reviewer #3: Yes

4. Have the authors made all data underlying the findings in their manuscript fully available?

Reviewer #1: Yes

Reviewer #2: Yes

Reviewer #3: Yes

5. Is the manuscript presented in an intelligible fashion and written in standard English?

Reviewer #1: Yes

Reviewer #2: Yes

Reviewer #3: Yes

6. Review Comments to the Author

Reviewer #1: (No Response)

Reviewer #2: the author addressed all the comments and th ems is now suitable for publication in PLos One Journal.

Reviewer #3: I would like to congratulate authors for the great job performed in this revised version of the manuscript. Most of the reviewers’ comments/suggestions were adequately considered and incorporated in the text. The addition of the calculation of Cohen’s d (effect size) largely contributed to better understand the magnitude of each reported change. Furthermore, the inclusion of the limitation section along with future directions significative contributed to the increase of the manuscript quality. I have no further suggestions/questions.

7. PLOS authors have the option to publish the peer review history of their article (what does this mean?). If published, this will include your full peer review and any attached files.

Reviewer #1: Yes: Claudia Meirelles

Reviewer #2: Yes: Angelina Zanesco

Reviewer #3: Yes: Rafael Herling Lambertucci

---

## [Editor Report · Acceptance letter]

18 Jun 2020

PONE-D-19-33282R1 

Activated whole-body arginine pathway in high-active mice 

Dear Dr. Granados:

I'm pleased to inform you that your manuscript has been deemed suitable for publication in PLOS ONE. Congratulations! Your manuscript is now with our production department. 

Kind regards, 

on behalf of

Dr. François Blachier 

Academic Editor

PLOS ONE